# Estimation of lifetime survival and predictors of mortality among TB with HIV co-infected children after test and treat strategies launched in Northwest, Ethiopia, 2021; a multicentre historical follow-up study

Ermias Sisay Chanie[1]*, Getnet Asmare Gelaye[1], Tesfaye Yimer Tadesse[2], Dejen Getaneh feleke[1], Wubet Taklual Admas[3], Eshetie Molla Alemu[3], Melkalem Mamoye Azanaw[3], Sofonyas Abebaw Tiruneh[3], Alemayehu Digssie Gebremariam[3], Binyam Minuye Birhane[4], Wubet Alebachew Bayih[4], Getachew Aragie[1]

1 Department of Pediatric and Child health Nursing, College of Health Sciences, Debre Tabor University, Ethiopia, 2 Department of Pharmacology, College of Health Sciences, Debre Tabor University, Ethiopia, 3 Department of Public health, College of Health Sciences, Debre Tabor University, Ethiopia, 4 Department of Maternity and Neonatal health Nursing, College of Health Sciences, Debre Tabor University, Ethiopia

* ermisis1888@gmail.com

**Data Availability Statement:** All relevant data are within the paper and its S1 File, S1 Data.

## Abstract

### Introduction

In resource-limited settings, the mortality rate among tuberculosis and human Immunodeficiency virus co-infected children is higher. However, there is no adequate evidence in Ethiopia in general and in the study area in particular. Hence, this study aims to estimate lifetime survival and predictors of mortality among TB with HIV co-infected children after test and treat strategies launched in Northwest Ethiopia Hospitals, 2021.

### Methods

Institution-based historical follow-up study was conducted in Northwest Ethiopia Hospitals among 227 Tuberculosis and Human Immunodeficiency Virus co-infected children from March 1, 2014, to January 12, 2021. The data were entered into Epi info-7 and then exported to STATA version 14 for analysis. The log-rank test was used to estimate the curve difference of the predictor variables. Bivariable cox-proportional hazard models were employed for each predictor variable. Additionally, those variables having a p-value < 0.25 in bivariate analysis were fitted into a multivariable cox-proportional hazards model. P-value < 0.05 was used to declare significance associated with the dependent variable.

### Results

From a total of 227 TB and HIV co-infected children, 39 died during the follow-up period. The overall mortality rate was 3.7 (95% CI (confidence interval): 2.9–4.7) per 100 person-years with a total of 1063.2-year observations. Cotrimoxazole preventive therapy (CPT)

**Funding:** The author(s) received no specific funding for this work.

**Competing interests:** The authors have declared that no competing interests exist.

**Abbreviations:** AHR, Adjusted hazard ratio; AIDS, Acquired immunodeficiency disease; ART, Antiretroviral therapy; CD4, Cluster of differentiation 4; CI, Confidence interval; CPT, Cotrimoxazole prophylactic therapy; HGB, Hemoglobin; HIV, Human immunodeficiency virus; IPT, Isoniazid prophylactic therapy; TB, Tuberculosis; OI, Opportunistic infection; PYO, Person per year observation; WHO, World health organization.

non-users [Adjusted Hazarded Ratio (AHR) = 3.8 (95% CI: 1.64–8.86)], presence of treatment failure [AHR = 3.0 (95% CI: 1.14–78.17)], and Cluster of differentiation 4(CD4) count below threshold [AHR = 2.7 (95% CI: 1.21–6.45)] were significant predictors of mortality.

## Conclusion

In this study, the mortality rate among TB and HIV co-infected children was found to be very high. The risk of mortality among TB and HIV co-infected children was associated with treatment failure, CD4 count below the threshold, and cotrimoxazole preventive therapy non-users. Further research should conduct to assess and improve the quality of ART service in Northwest Ethiopia Hospitals.

## Introduction

Human immunodeficiency virus(HIV) and TB(Tuberculosis) are infectious diseases that usually occur together with each worsening one another [1]. HIV is a virus that attacks the body's immune system CD4 cells. Over time, untreated HIV reduces the number of CD4 cells in the body that the body can't fight off infections [2]. HIV is increasing the risk of acquiring TB infection, and TB enhances HIV replication by accelerating the natural evolution of HIV infection [3]. Globally, TB is the leading cause of death from a single infectious agent [4], including persons living with HIV [5].

Globally, there were 10 million people with TB in 2020, 1.2 million deaths from HIV-negative people and an additional 208,000 deaths from HIV-positive individuals [6]. Nowadays, the prevalence of TB/HIV co-infection has been increased and become a major public health concern in worldwide, Indeed the highest cases were reported in Sub-Saharan African countries [7]. TB and HIV co-infections are often called a twin epidemic [8], one to four HIV-related death is a result of TB in children [9], and TB carries significant mortality regardless of the ART and adequate anti-tuberculosis treatment in these children [10].

In fact, children living with HIV are highly vulnerable to mortality than adults, due to a very young age rely on their parents/caregivers to access health care services [11]. worldwide in 2018, 251,000 people who had both TB /HIV are estimated to have died. Of these, 211,000 TB/HIV death from Africa, and 30,000 were children (male, 0-14(16,000)) and (Female 0-14 (14,000)) [12], and almost all from sub-Saharan Africa [13]. Indeed, TB has the lion's share in HIV-related deaths [8].

HIV and TB are the leading causes of death from infectious diseases worldwide in HIV-infected children [14], and they are major public health concerns particularly in resource-limited countries like Ethiopia.

HIV/TB co-infection mortality in Sub-Saharan countries continues taking the leading position and heavily affected [15], and Ethiopia ranked seventh among the 22 heavily affected countries [3]. For this reason, data on HIV/TB co-infection in children are still lacking in resource-limited settings [16], and TB -HIV-associated mortality often remains unascertained [17].

Africa countries including Ethiopia adopted the WHO "Global TB program in 2015" launched to achieve the Sustainable Development Goals (SDG) and END-TB strategies, which targeted to decrease TB death and incidence by 90% and 80% respectively by 2030 through joint efforts to the burden of the two epidemics [18]. There are efforts at a global and national level that have been done to achieve the Sustainable Development Goals (SDG) and END-TB

strategies. However, TB accounts for approximately 40% of facility-based HIV/AIDS-related deaths in the resource-limited settings among TB/HIV co-infected children [17], and Ethiopia is one of them [3]. Besides, the burden of TB/HIV co-infection is still intolerable in Ethiopia in general and in the study area in particular, and it attracts the attention of the government and researchers. Hence, this study aims to estimate lifetime survival and predictors of mortality among TB with HIV co-infected children after test and treat strategies launched in northwest Ethiopia hospitals, 2021.

## Methods and materials

### Study settings and subjects

A historical follow-up study was conducted from March 1, 2014, to January 12, 2021, in Northwest Ethiopia Hospitals, The Hospitals namely Debre Tabor Compressive Specialized Hospital, Gondar Compressive Specialized Hospital, and South Gondar primary Hospitals. ART service is one of the services delivered in the Hospitals by ART case-team includes physicians, nurses, pharmacists, laboratory technicians who took comprehensive ART training. A total of 239 TB HIV co-infected children were enrolled during the study period in the above hospitals. All TB and HIV co-infected children in the above hospitals from March 1, 2014, to January 12, 2021, were included. Whereas TB and HIV co-infected children incomplete baseline information TB status and the outcome variable(mortality) were excluded.

### Sample size determination

The sample size was calculated by using Log-rank survival data analysis of the two-population proportion formula based on the following important assumptions- 95% confidence level, 80% optimum statistical power, and taking type one error 5%. By considering a study was conducted in eastern Ethiopia [19], by taking sex as a predictor variable (on the male as the exposed group denoted by q1 (0.38) and female group denoted by q0 (0.53), and then the total sample size, after adding 5% as incomplete medical records, and the final sample size was 372. However, all TB and HIV co-infected children from March 1, 2014, to January 12, 2021, were 239. Hence, all study participants were included in the study.

### Data collection tools and procedures

The data extraction tool was developed from the patient registry book prepared by the Ethiopian Federal Ministry of Health. Data were collected from patient ART record cards and registers from March 1, 2014, to January 12, 2021. The data extraction tool contained socio-demographic, clinical, and treatment-related information. The data were collected by six BSc Nurse and supervised by two MSc in paediatrics and child health Nursing practitioners. A pre-test was conducted among 5% of the sample size of medical records at Debre Tabor Compressive Specialized Hospital to check the completeness of the medical records. Two-days training was given about data collection procedures and supervisions.

### Operational definition

**Time to death.** The time from TB/HIV co-infection to the occurrence of the event (i.e., death) during the follow-up period.

**Censored.** A child was considered as censored if the child is lost to follow-up or transfer out to another service or if the child was alive until the end of the study period.

**Test-and-treat.** Is an intervention strategy in which the population at risk is screened for HIV infection and diagnosed HIV infected individuals receive early treatment, aiming to

eliminate HIV as it reduces the rate of spreading the virus to other people. In Ethiopia test and treat strategies among HIV infected children were launched since in 2014 G.C.

**A CD4 count.** CD4 below the threshold level was classified based on the age of the child's (i.e. infants CD4<1500/mm3, 12–35 months <750/mm3, 36–59 months <350/mm3 and ≥5 years <200/mm3) [20].

**Underweight or stunting.** Was defined as weight for age Z-score < −2 SD for under-five children and BMI for age Z-score < −2 SD for older children [3].

**Anemia.** Was defined as having a hemoglobin level ≤ of 10 mg/dl [20].

**Adherence to ART.** Was classified based on the percentage of drug dosage calculated from the total monthly doses of ART drugs. (Good >95%, fair 85–94%, and poor <85%) [7].

## Data processing and analysis

The data were entered into Epi info -7 and then exported to STATA version 14 for analysis. The descriptive statics were explored through tables and graphs. The mortality rate was calculated by dividing the number of children who died during the follow-up period by the Child-Years of follow-up. Kaplan Meier curve was used to estimate the survival time. Besides, the log-rank test was used to estimate the curve difference of the predictor variables. The required assumption of the Cox-proportional hazard regression model was checked through Schoenfeld residual ph test and Log ph plot. Bivariable cox-proportional hazard models were employed for each predictor variable. Additionally, those variables having p-value < 0.25 in bivariate analysis were fitted into a multivariable cox-proportional hazards model. P-value < 0.05 was used to declare a significant association with the dependent variable.

**Ethics approval and consent to participate.** Ethical clearance was granted from the Institutional Review Board (IRB) of Debre Tabor University Ethical Review Committee with Ref No 842/21. Besides, a permission letter was obtained from each hospital administrators. The informed written consent was obtained from the ART focal person in each hospital on behalf of the study participants. Since it is secondary data. A total of 239 patient records were used in this retrospective study, including the date range (4–167 months) in which medical records were accessed during the follow-up period.

## Results

### Socio-demographic characteristics

A total of 239 TB/HIV co-infected children enrolled on ART during the follow-up period, 12 were excluded due to incomplete data. Of 227 TB/HIV co-infected children, nearly half of 117 (51.54% and 122(53.74%) age between 6.10 years and males respectively. Besides, nearly half of 125 (55.07%) of the caregivers of the child were housewife, whereas 100 (44.05%) of the caregivers were unable to read and write. The majority 187 (82.38%) and 204 (89.87%) of children were urban residence and livening with their parents in the follow-up period respective (Table 1).

### Clinical and treatment-related characteristics

Out of 227 TB/HIV co-infected children, 66 (29.07%) were initiated ART with EFV regimen based. From the total children, 103 (45.37%) had initial regimen change. Of these, 43 (41.75%) change their initial regimen due to treatment failure. Besides, 68 (29.96%) of TB/HIV co-infected children were reported different kinds of drug side effects.

From the total 277 TB/HIV co-infected children, the majority 183 (80.62%) of them were taken CPT, while less proportion 152 (66.96%) were taking IPT. A large portion of TB/HIV

**Table 1. Socio-demographic characteristics among TB and HIV co-infected children after test and treat strategies launched in northwest Ethiopia hospitals, 2021.**

| Characteristics | | Frequency | Percent |
|---|---|---|---|
| Age (years) | < 1 | 3 | 1.32 |
| | 1–5 | 61 | 26.87 |
| | 6–10 | 117 | 51.54 |
| | >11 | 46 | 20.26 |
| Sex | Male | 122 | 53.74 |
| | Female | 105 | 46.26 |
| Residence | Urban | 187 | 82.38 |
| | Rural | 40 | 17.62 |
| Child lives Orphaned | Yes | 23 | 10.13 |
| | No | 204 | 89.87 |
| Caregiver's occupational status | Housewife | 125 | 55.07 |
| | Governmental employee | 50 | 22.03 |
| | Non-governmental employee | 23 | 10.13 |
| | Merchant | 29 | 12.78 |
| Caregivers of educational status | Unable | 100 | 44.05 |
| | Primary | 68 | 29.96 |
| | Secondary and above | 59 | 25.99 |

co-infected children, 177 (77.97%), 186 (81.94%),161 (70.93%), and 169 (74.45%) were CD4 counts above the threshold level, Hgb level > = 10 mg/dl, WHO stage III, a good level of adherence to ART during the follow-up period respectively. On the other hand, nearly half of children, 118 (51.98%), 99 (43.61%), and 103 (45.37%) of TB/HIV co-infected children were underweight, stunting, and had an opportunistic infection in the follow-up period respectively. From the 227 TB/HIV co-infected children, 98 (43.17%) were followed for more than 60 months (Table 2).

## Kaplan-Meier survival curve

From a total of 227 TB/HIV co-infected children, 39 were died, which a proportion of 17.2% during the follow-up period (Fig 1).

The total mortality rate among TB/HIV co-infected children was 3.7 (95%CI: 2.9–4.7) per 100 person-years. The children followed with the range from 04 to 167 months days, which yields a total of 12,758 months or 1,063.17 years at risk (Fig 2).

## Predictors of mortality

In bivariate cox proportional hazard model, residence, occupation of the mother, residence, a child lives orphaned, caregiver's occupational status, caregivers of educational status, initial ART regimen, initial regiment change, treatment failure, duration on ART, IPT, CPT, CD4 counts or % level, Hgb level, WHO stage, history of OI, and level of adherence to ART of variable had P-value less than or equal to 0.25 and entered into for multivariate cox proportional hazard.

In the multivariate cox proportional hazard model, treatment failure, CPT non-users, and CD4 counts below threshold level were significant predictors of mortality among TB/HIV co-infected children.

The hazards of mortality in children with a CD4 count below threshold level were 2.7 times higher than those children with a CD4 count above threshold level [AHR: 2.7(95% CI: 1.21, 6.45)].

**Table 2. Clinical and treatment-related characteristics among TB and HIV co-infected children after test and treat strategies launched in northwest Ethiopia hospitals, 2021.**

| Characteristics | | Frequency | Percent |
|---|---|---|---|
| Initial ART regiments based on NNRTIs | EFV-based | 66 | 29.07 |
| | NVP, PI, and others | 161 | 70.93 |
| Initial regiment change | Yes | 103 | 45.37 |
| | No | 124 | 54.63 |
| Reason for regiment change(n = 103) | Side effect/toxicities | 21 | 20.39 |
| | Stockout | 39 | 37.86 |
| | Treatment failure | 43 | 41.75 |
| Treatment failure | Yes | 43 | 18.94 |
| | No | 184 | 81.06 |
| Drug side effect | Yes | 68 | 29.96 |
| | No | 159 | 70.04 |
| Duration On ART | <60 months | 129 | 56.83 |
| | ≥60 months | 98 | 43.17 |
| Isoniazid | Yes | 152 | 66.96 |
| | No | 75 | 33.04 |
| Co-trimoxazole preventive therapy | Yes | 183 | 80.62 |
| | No | 44 | 19.38 |
| CD4 counts or % level | Below threshold | 50 | 22.03 |
| | Above threshold | 177 | 77.97 |
| HGB level | < 10 mg/dl | 41 | 18.06 |
| | > = 10 mg/dl | 186 | 81.94 |
| WHO stage | Stage III | 161 | 70.93 |
| | Stage IV | 66 | 29.07 |
| Opportunistic infections | Yes | 103 | 45.37 |
| | No | 124 | 54.63 |
| ART adherence | Good | 169 | 74.45 |
| | Poor/Fair | 58 | 25.55 |
| Height for age | Stunting | 99 | 43.61 |
| | Normal | 128 | 56.39 |
| Weight for age | Underweight | 118 | 51.98 |
| | Normal | 109 | 48.02 |

NNRTIs = Non-Nucleoside Reverse Transcriptase Inhibitors, EFV = Efavirenz, NVP = Nevirapine, PI = Protease Inhibitor.

The hazards of mortality in children with treatment failure were 3.0 times higher than those children without treatment failure [AHR: 3.0(95% CI: 1.14, 8.17)].

The hazards of mortality in children with CPT non-users were 3.8 times higher than those children with CPT users [AHR: 3.8(95% CI: 1.64, 8.86)] (Table 3). Additionally, the log-rank test of the between the categories variable of the predictors was estimated (Figs 3–5).

## Discussion

In this study, the overall mortality rate among TB HIV co-infected children in Northwest Ethiopia Hospitals was found to be 3.7 (95% CI: 2.9–4.7) per 100 person-years. Additionally, the proportion of mortality was found to be 17.2%. This finding is comparable with the study conducted in Gondar Ethiopia was 3.27 PPY [20], in Tanzania was 17.5% [21], and in Thailand was 17% [22]. However, the finding of this study was higher than the studies conducted in

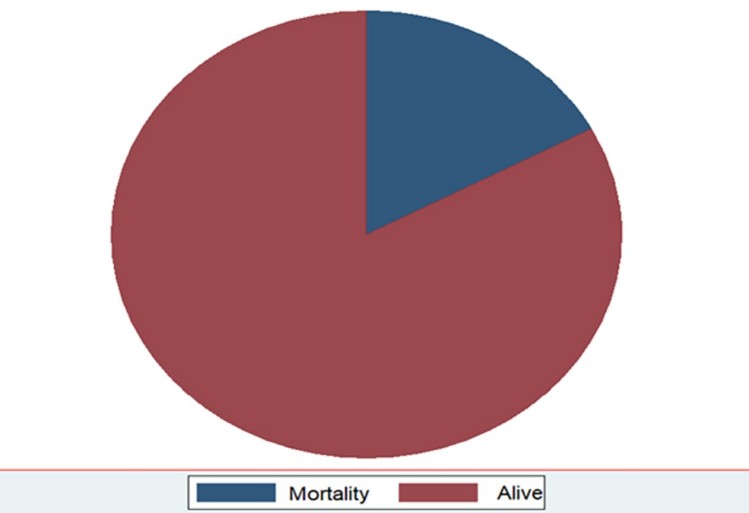

**Fig 1. Proportion of mortality among TB and HIV co-infected children by CD4 counts below the threshold level in northwest Ethiopia hospitals, 2021.**

seven referral Hospitals in Ethiopia was 14% [19], in Uganda was 10.47% (38), in Nigeria was 1.4 PPY [23], in South Africa was 12.4% [24], in Botswana was 13.6% [25], and the Democratic Republic of the Congo was 13% [26]. On the other hand, the finding of this study is lower than the study conducted in Jimma, Ethiopia was 20.2% [27].

The possible justification might be that developed/developing countries have better quick tests or screening for TB and/or HIV infection than resource-limited setting countries such as Ethiopia that can help the patient treatment in a timely and then to improve. Moreover, there is a constraint of knowledge and skill of the health care providers to screen, diagnose, and treat the diseases of TB/HIV co-infection and its related complication in Africa [28, 29]. Even the healthcare providers have adequate knowledge, there is resource limitation to combat TB/HIV co-infection.

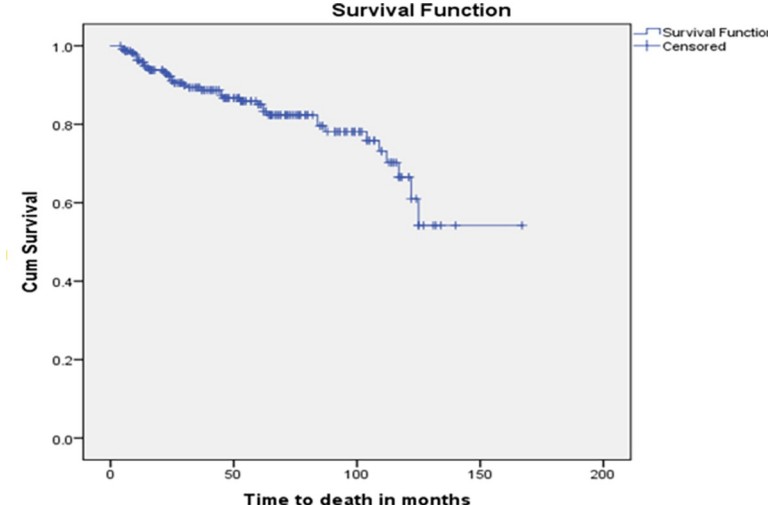

**Fig 2. Kaplan-Meier of survival curve among TB and HIV co-infected children after test and treat strategies launched in northwest Ethiopia hospitals, 2021.**

**Table 3. Bivariable and multivariable Cox-regression of predictor variable among TB and HIV co-infected children after test and treat strategies in northwest Ethiopia hospitals, 2021.**

| Characteristics | | Survival status | | CHR (95% CI) | AHR (95% CI) | P-Value |
|---|---|---|---|---|---|---|
| | | No | Yes | | | |
| Age (years) | < 1 | 1 | 2 | 2.7(0.33–21.5) | - | |
| | 1–5 | 9 | 52 | 1.1(0.41–2.78) | - | |
| | 6–10 | 21 | 96 | 1.1(0.47–2.41) | - | |
| | >11 | 8 | 38 | Ref | | |
| Sex | Male | 18 | 104 | 0.8(0.42–1.51) | - | |
| | Female | 21 | 84 | Ref | | |
| Residence | Urban | 30 | 157 | Ref | Ref | |
| | Rural | 9 | 31 | 1.9(0.88–4.01) | 1.7(0.70–5.51) | 0.248 |
| Child lives Orphaned | Yes | 9 | 14 | 2.5(1.18–5.25) | 1.5(0.63–3.59) | 0.351 |
| | No | 30 | 174 | Ref | Ref | |
| Caregiver's occupational status | Housewife | 21 | 104 | 1.3(0.44–3.78) | - | |
| | Governmental employee | 6 | 44 | 0.8(0.24–2.97) | - | |
| | Non-governmental employee | 8 | 15 | 2.5(0.75–8.31) | - | |
| | Merchant | 4 | 25 | Ref | | |
| Caregiver's of educational status | Unable | 20 | 80 | 1.2(0.57–2.61) | - | |
| | Primary | 9 | 59 | 0.8(0.30–1.85) | - | |
| | Secondary and above | 10 | 49 | Ref | | |
| ART adherence | Good | 21 | 148 | Ref | Ref | |
| | Poor/Fair | 18 | 40 | 4.0(2.14–7.60) | 1.6(0.75–3.41) | 0.228 |
| Height for age | Stunting | 18 | 81 | 1.1(0.59–2.08) | | |
| | Normal | 21 | 107 | Ref | | |
| Weight for age | Underweight | 22 | 96 | 1.4(0.74–2.63) | | |
| | Normal | 17 | 92 | Ref | | |
| Initial ART regimen | EFV-based | 20 | 46 | 2.5(1.35–4.76) | 1.1(0.52–2.49) | 0.744 |
| | NVP, PI and others based | 19 | 142 | Ref | Ref | |
| Initial regiment change | Yes | 28 | 75 | 4.0(1.99–8.17) | 1.1(0.41–3.12) | 0.804 |
| | No | 11 | 113 | Ref | Ref | |
| Reason for regiment change(n = 103) | Treatment failure | 21 | 22 | 2.1(0.7–16.7) | - | |
| | Side effect/toxicities | 3 | 18 | 0.7(0.15–3.11) | - | |
| | Stockout | 4 | 35 | Ref | | |
| Drug side effect | Yes | 13 | 55 | 1.2(0.64–2.43) | - | |
| | No | 26 | 133 | Ref | | |
| Treatment failure | Yes | 21 | 22 | **8.6(4.50–16.4)** | **3.0(1.14–8.17)** | **0.031***|
| | No | 18 | 166 | Ref | Ref | |
| IPT | Yes | 34 | 118 | 3.4(1.33–8.80) | 2.0(0.71–5.51) | 0.194 |
| | No | 5 | 70 | Ref | Ref | |
| CPT | No | 25 | 19 | **10.7(5.5–20.7)** | **3.8(1.64–8.86)** | **0.002**** |
| | Yes | 14 | 169 | Ref | Ref | |
| CD4 counts or % level | Below threshold | 25 | 25 | **7.4(3.85–14.4)** | **2.7(1.21–6.45)** | **0.026***|
| | Above threshold | 14 | 163 | Ref | Ref | |
| HGB level | < 10 mg/dl | 21 | 20 | 7.0(0.37–13.1) | 1.7(0.35–2.37) | 0.856 |
| | > = 10 mg/dl | 18 | 168 | Ref | Ref | |
| WHO stage | Stage III | 16 | 145 | Ref | Ref | |
| | Stage IV | 23 | 43 | 4.5(2.39–8.63) | 1.9(0.9–2.7) | 0.791 |

(*Continued*)

**Table 3.** (Continued)

| Characteristics | | Survival status | | CHR (95% CI) | AHR (95% CI) | P-Value |
|---|---|---|---|---|---|---|
| | | No | Yes | | | |
| Opportunistic infection | Yes | 29 | 74 | 3.9(1.92–8.07) | 1.8(0.71–4.53) | 0.213 |
| | No | 10 | 114 | Ref | Ref | |

*Significant at <0.05

** Significant at <0.01; CHR = Crude hazard ratio; AHR = adjusted hazard ratio; Ref = reference category; CI = confidence interval.

The hazards of mortality in children with a CD4 count below the threshold level were 2.7 times higher than those children with a CD4 count above the threshold level. This finding is supported by the studies conducted in another setting [7, 19, 26, 27, 30]. In fact, patients with depilation of CD4 count can develop more severe disease or advanced opportunistic infections including pneumocystis pneumonia, diarrhoea, oral/oesophagal candidiasis which leads to compilation and death [31, 32].

The hazards of mortality in children with treatment failure were 3.0 times higher than those children without treatment failure. This finding is supported by the studies conducted in another setting [30]. Generally, children not on effective HIV treatment in overtime, then HIV has done a lot of damage to the immune system. Since taking HIV medicines as prescribed can help keep the viral suppression and your CD4 cell count high [33, 34]. Moreover, treatment failure worsens the risk of mortality of the patient as a result of ARV combination and treatment duration, which implies the antiretroviral drugs are no longer able to suppress the virus or prevent the deterioration of your immune system [35, 36].

The hazards of mortality in children with CPT non-users were 3.8 times higher than those children with CPT users. This finding is supported by the studies conducted in another setting (25,31,33). Actually, cotrimoxazole prophylaxis is given for HIV-infected children to avoid either the first occurrence or their recurrence of opportunistic infection that can increase the survival of the children. On the other hand, children who had co-trimoxazole non-user will face several problems such as treatment failure, CD4 count depletion, and the occurrence of opportunistic infection that could lead to death [37, 38]. This study does have inherent limitations due to the retrospective nature of the study. Also, some of the important predictors

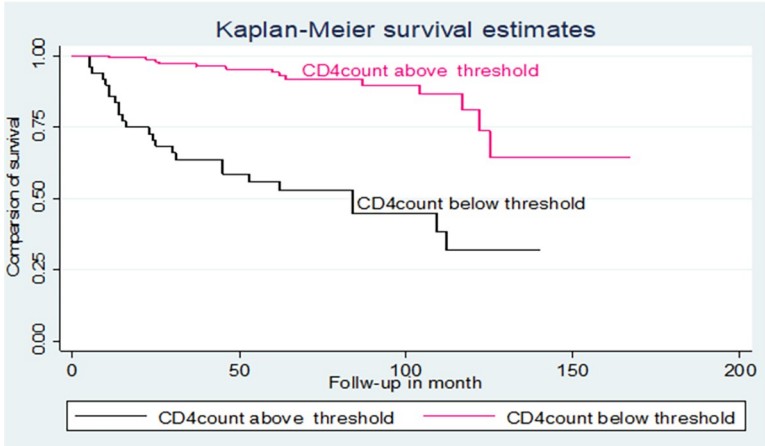

**Fig 3. Kaplan-Meier of survival curve of among TB and HIV co-infected children by CD4 counts below the threshold level in northwest Ethiopia hospitals, 2021.**

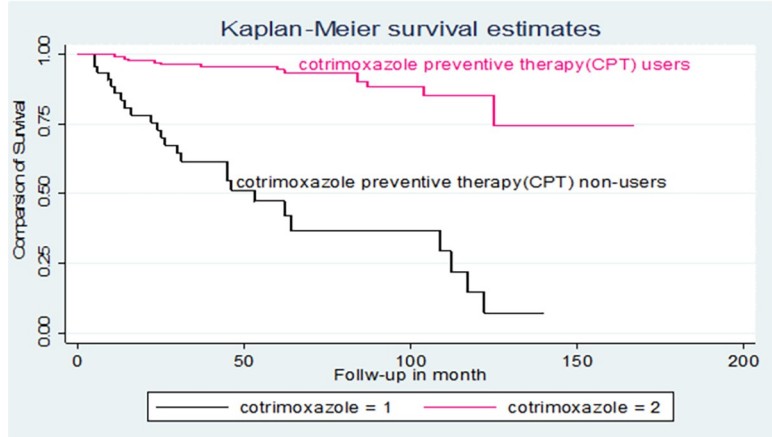

**Fig 4. Kaplan-Meier of survival curve of among TB and HIV co-infected children CPT non-users in northwest Ethiopia hospitals, 2021.**

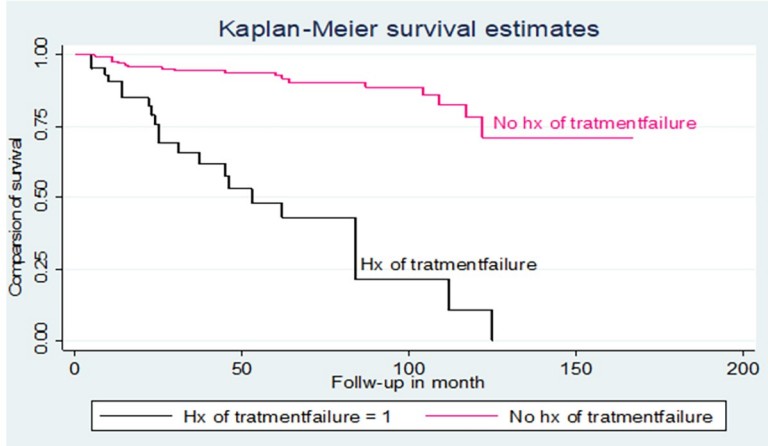

**Fig 5. Kaplan-Meier of survival curve of among TB and HIV co-infected children by treatment failure in northwest Ethiopia hospitals, 2021.**

might have a significant association with mortality but were not investigated due to unavailability in the medical or ART registers logbook.

## Conclusion

In this study, the mortality rate among TB and HIV co-infected children was found to be very high. The risk of mortality among TB and HIV co-infected children was associated with treatment failure, CD4 count below the threshold, and cotrimoxazole preventive therapy non-users. Further research should conduct to assess and improve the quality of ART service in Northwest Ethiopia Hospitals.

## Supporting information

**S1 Data.**
(DTA)

**S1 File.**
(DOCX)

## Acknowledgments

Firstly, we would like to express our deepest gratitude to Debre tabor university, secondly the ART focal person of each South Gondar hospital, and lastly our heartfelt also goes to all individuals who participated in the study and data collectors.

## Author Contributions

**Conceptualization:** Ermias Sisay Chanie, Getnet Asmare Gelaye, Getachew Aragie.

**Data curation:** Ermias Sisay Chanie, Tesfaye Yimer Tadesse, Dejen Getaneh feleke, Wubet Taklual Admas, Eshetie Molla Alemu, Sofonyas Abebaw Tiruneh, Alemayehu Digssie Gebremariam, Binyam Minuye Birhane, Wubet Alebachew Bayih, Getachew Aragie.

**Formal analysis:** Ermias Sisay Chanie, Melkalem Mamoye Azanaw, Binyam Minuye Birhane.

**Investigation:** Ermias Sisay Chanie, Getnet Asmare Gelaye, Tesfaye Yimer Tadesse, Dejen Getaneh feleke, Wubet Taklual Admas, Eshetie Molla Alemu, Melkalem Mamoye Azanaw, Sofonyas Abebaw Tiruneh, Alemayehu Digssie Gebremariam, Wubet Alebachew Bayih, Getachew Aragie.

**Methodology:** Ermias Sisay Chanie, Sofonyas Abebaw Tiruneh.

**Project administration:** Ermias Sisay Chanie.

**Resources:** Ermias Sisay Chanie.

**Software:** Ermias Sisay Chanie, Wubet Alebachew Bayih, Getachew Aragie.

**Supervision:** Ermias Sisay Chanie, Dejen Getaneh feleke, Wubet Alebachew Bayih, Getachew Aragie.

**Validation:** Ermias Sisay Chanie, Wubet Taklual Admas, Alemayehu Digssie Gebremariam, Binyam Minuye Birhane.

**Visualization:** Ermias Sisay Chanie, Melkalem Mamoye Azanaw, Alemayehu Digssie Gebremariam.

**Writing – original draft:** Ermias Sisay Chanie, Getnet Asmare Gelaye.

**Writing – review & editing:** Tesfaye Yimer Tadesse, Dejen Getaneh feleke, Wubet Taklual Admas, Eshetie Molla Alemu, Melkalem Mamoye Azanaw, Sofonyas Abebaw Tiruneh, Binyam Minuye Birhane, Wubet Alebachew Bayih, Getachew Aragie.

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
