## [Decision Letter · Decision Letter 0]

4 May 2021

PONE-D-21-04696

Estimation of Lifetime Survival and Predictors of Mortality Among TB with HIV Co-infected Children After Test and Treat Strategies Launched in South Gondar Public Hospitals, Northwest, Ethiopia, 2021; A Multicentre Historical Follow-up Study

PLOS ONE

Dear Dr. Ermias Sisay Chanie,

Thank you for submitting your manuscript to PLOS ONE. After careful consideration, we feel that it has merit but does not fully meet PLOS ONE’s publication criteria as it currently stands. Therefore, we invite you to submit a revised version of the manuscript that addresses the points raised during the review process.

We look forward to receiving your revised manuscript.

Kind regards,

Qigui Yu, M.D./Ph.D

Academic Editor

PLOS ONE

Additional Editor Comments:

Suggest to get help from someone with full professional proficiency in English to revise the manuscript.

2. In your ethics statement in the manuscript and in the online submission form, please ensure that you have discussed whether all data/samples were fully anonymized before you accessed them and/or whether the IRB or ethics committee waived the requirement for informed consent. If patients provided informed written consent to have data/samples from their medical records used in research, please include this information. Please also include the date that ethics approval was granted.

3. In the ethics statement in the manuscript and in the online submission form, please provide additional information about the patient records/samples used in your retrospective study, including the date range (month and year) during which patients' medical records/samples were accessed.

Reviewers' comments:

Reviewer's Responses to Questions

**Comments to the Author**

1. Is the manuscript technically sound, and do the data support the conclusions?

Reviewer #1: Yes

Reviewer #2: No

2. Has the statistical analysis been performed appropriately and rigorously? 

Reviewer #1: Yes

Reviewer #2: No

3. Have the authors made all data underlying the findings in their manuscript fully available?

Reviewer #1: Yes

Reviewer #2: Yes

4. Is the manuscript presented in an intelligible fashion and written in standard English?

Reviewer #1: No

Reviewer #2: No

5. Review Comments to the Author

Reviewer #1: In this study, the authors performed institution-based retrospective follow-up study conducted from March 1, 2014, to January 12, 2021 on survival and predictors of mortality among 227 TB and HIV co-infected children in Ethiopia. They found that this population of children had a high mortality rate of 17% and that cotrimoxazole preventive therapy (CPT) non-users, ART treatment failure, and CD4 count below threshold were significant predictors of mortality. The study was well done, but the writing needs substantial improvement. There are no other major concerns. The following are some minor concerns and questions.

1) Please provide explanation to the test and treat strategies and ART service

2) Does duration on ART mean that the following time of the patients during the study.

3) How many patients in Stage III & IV were in Stage IV? Would Stage IV be a predictor of mortality?

4) In Table 3 CPT, the Yes and No groups were mislabeled.

5) The Figure legends for Kaplan-Meier survival curves of CD4 counts are mixed-up. The legends for CPT user and non-user were the same.

6) What were the major causes of death?

Reviewer #2: the manuscript "Estimation of Lifetime Survival and Predictors of Mortality Among TB with HIV Co-infected

Children After Test and Treat Strategies Launched in South Gondar Public Hospitals, Northwest,

Ethiopia, 2021; A Multicentre Historical Follow-up Study" by Chanie et al touched a very important heath issue among the HIV/TB co-infected people. However, the studies are not very organized so its enthusiasm is reduced.

Major concerns:

1. language: The whole paper is very hard to understand, current wiring is very sloppy.

2. Statistical analysis is lacking for the Tables 1 and 2. Although the statistical power (p value) was shown for the Table 3, it lacks a description of how the authors did the statistical analysis.

3. As the authors concluded "In this study, mortality among TB and HIV co-infected children was found to be very high. Treatment failure and CD4 count below the threshold were the independent predictors of mortality.

Hence, it is better a special emphasis on monitoring and managing regularly of these contributing

factors. Besides, strengthen the WHO recommendation of cotrimoxazole preventive therapy for all

patients living with HIV is crucial." it is not novel at all, e.g. people have known the importance of CD4 count.

suggest the authors making Bar graphs for Table 1 and Table 2 to show more detailed data in clearer manner.

6. PLOS authors have the option to publish the peer review history of their article (what does this mean?). If published, this will include your full peer review and any attached files.

Reviewer #1: No

Reviewer #2: No

---

## [Author Response · Author response to Decision Letter 0]

21 May 2021

Dear professor Qigui Yu (PLOS ONE journal chief Editor) 

After going through the entire manuscript entitled "Estimation of Lifetime Survival and Predictors of Mortality Among TB with HIV Co-infected Children After Test and Treat Strategies Launched in South Gondar Public Hospitals, Northwest, Ethiopia, 2021; A Multicentre Historical Follow-up Study" you forwarded your constructive comments which we missed to touch. Therefore, we are glad enough to express our sincerest thanks for your constructive Editors and reviewers’ comment that could help improve novelty of our effort. Additionally, those comments are all valuable and very helpful for revising and improving our paper, as well as an important guiding to significance the article. 

We hope you find our manuscript suitable for publication and look forward hearing from you. It is needless to state that we are happy to provide any more details in support of this manuscript. Hence, we are living in resource limited setting that unable access the international English language experts to write very clearly. The main corrections in the paper and the responds to the Editor’s and reviewer’s comments are as flowing. The revised portion are marked in the yellow highlighted in revised version manuscript.

---

## [Decision Letter · Decision Letter 1]

22 Jul 2021

PONE-D-21-04696R1

Estimation of Lifetime Survival and Predictors of Mortality Among TB with HIV Co-infected Children After Test and Treat Strategies Launched in Northwest, Ethiopia, 2021; A Multicentre Historical Follow-up Study

PLOS ONE

Dear Ermias Sisay Chanie,

Thank you for submitting your manuscript to PLOS ONE. After careful consideration, we feel that it has merit but does not fully meet PLOS ONE’s publication criteria as it currently stands. Therefore, we invite you to submit a revised version of the manuscript that addresses the points raised during the review process.

We look forward to receiving your revised manuscript.

Kind regards,

Qigui Yu, M.D./Ph.D

Academic Editor

PLOS ONE

Journal Requirements:

Reviewers' comments:

Reviewer's Responses to Questions

**Comments to the Author**

1. If the authors have adequately addressed your comments raised in a previous round of review and you feel that this manuscript is now acceptable for publication, you may indicate that here to bypass the “Comments to the Author” section, enter your conflict of interest statement in the “Confidential to Editor” section, and submit your "Accept" recommendation.

Reviewer #1: (No Response)

Reviewer #3: (No Response)

2. Is the manuscript technically sound, and do the data support the conclusions?

Reviewer #1: Yes

Reviewer #3: Partly

3. Has the statistical analysis been performed appropriately and rigorously? 

Reviewer #1: Yes

Reviewer #3: Yes

4. Have the authors made all data underlying the findings in their manuscript fully available?

Reviewer #1: Yes

Reviewer #3: Yes

5. Is the manuscript presented in an intelligible fashion and written in standard English?

Reviewer #1: No

Reviewer #3: Yes

6. Review Comments to the Author

Reviewer #1: The major issue had been poor English writing, which was improved in this revised version. However, there are still many grammar mistakes and the overall flow of the sentences need to be fixed. There are also some misstatements and typos, some of which are listed below.

The statement in the Abstract “However, there is no prior evidence in Ethiopia.” is not accurate as the authors themselves cited a reference to show the opposite (ref #18, Survival and predictors of mortality among children co-infected with tuberculosis and human immunodeficiency virus at University of Gondar Comprehensive Specialized Hospital, Northwest Ethiopia. A retrospective follow-up study. PLoS ONE [Internet]. 2018 May 22”). It appears that this current study and the above-mentioned 2018 study both recruited patients from the same Gondar Comprehensive Specialized Hospital. Were there any patients included in both studies? Some discussion on the similarities and differences between both studies would be helpful.

As the risk of mortality is associated with CPT therapy non-users, the concluding should emphasize CPT use along with ART therapy.

Some of the references need to be updated. For example, World Health Organization. Global tuberculosis report 2020 is available (ref #4); The data shown in the sentences “In 2015 worldwide, 1.6 million people…” should be replaced with more recent data.

Should the overall mortality rate be 3.7 per 100 person-years instead of 3.7 per person-years.

Should Debre Tabor compressive specialized hospital be Debre Tabor Comprehensive Specialized Hospital?

In “by taking sex as a predictor variable (on the male as the exposed group

denoted by q1 (0.38) and female group denoted by q0 (0.53)”, where the numbers come from?

The sentence “The patients provided informed written consent to have data/samples from their medical records used in research.” needs re-wording as the patients were too young to provide such consent.

In “half of 117 (51.54% and 122(53.74%) male and age between 6.10 years respectively.”, the order was wrong: 122 is for male and 117 for the age group.

In “118 (51.98%), 99 (43.61%), 103 (45.37%), and 218 (77.03%)…), what 218 is for?

Reviewer #3: The authors studied lifetime survival and predictors of mortality in children with TB and HIV coinfection. Overall mortality rate was 3.7, which as associated with treatment failure, especially CD4 count below the threshold. The manuscript has improved adequately, but there are minor concerns that should be made clear for the conclusion in relation to treatment failure.

1. Antiretroviral drug (ARV) combination in the patient cohorts and groups, especially those with treatment failure, since 18.94% co-infected children showed treatment failure;

2. Did the ART cessation happen in the co-infected patients, which likely contribute to treatment failure, such as CD4 decline to the levels of threshold and emergence of opportunistic infection. These are important to make statistical analysis and conclusion;

3. The ARV combination and treatment duration in cohort with treatment failure should be considered in the group classification in the statistical analysis, at least discussed.

7. PLOS authors have the option to publish the peer review history of their article (what does this mean?). If published, this will include your full peer review and any attached files.

Reviewer #1: No

Reviewer #3: No

---

## [Author Response · Author response to Decision Letter 1]

24 Jul 2021

Reviewer#1:

Comments #1: The major issue had been poor English writing, which was improved in this revised version. However, there are still many grammar mistakes and the overall flow of the sentences need to be fixed. There are also some misstatements and typos, some of which are listed below.

Author’s response: Undoubtedly! It is modified based on the given comment. 

After frequent proofreading of the manuscript had several grammatical wordings and spelling errors. Hence, based on the given comment revision was made. These changes are found throughout the manuscript. These changes are found throughout the revised version manuscript

Comments #2: The statement in the Abstract “However, there is no prior evidence in Ethiopia.” is not accurate as the authors themselves cited a reference to show the opposite (ref #18, Survival and predictors of mortality among children co-infected with tuberculosis and human immunodeficiency virus at University of Gondar Comprehensive Specialized Hospital, Northwest Ethiopia. A retrospective follow-up study. PLoS ONE [Internet]. 2018 May 22”). It appears that this current study and the above-mentioned 2018 study both recruited patients from the same Gondar Comprehensive Specialized Hospital. Were there any patients included in both studies? Some discussion on the similarities and differences between both studies would be helpful. As the risk of mortality is associated with CPT therapy non-users, the conclusion should emphasize CPT use along with ART therapy

Author’s response: We strongly agree with the essence of incorporating this comment!

Sure, we are stated that there is no prior evidence in Ethiopia regarding to estimation of lifetime survival and predictors of mortality among TB with HIV co-infected children after test and treat strategies launched in Northwest, Ethiopia, 2021. What makes this research differ from the previous one (i.e., ref #18 from 2005-2018) is: This study was conducted in after test and treat strategies launched (i.e., after 20014), recent evidence, and multicenter. However, we are concluded to revised as limited evidence than no evidence. 

Comments #3: Some of the references need to be updated. For example, World Health Organization. Global tuberculosis report 2020 is available (ref #4); The data shown in the sentences “In 2015 worldwide, 1.6 million people…” should be replaced with more recent data.

Author’s response: Unquestionably! It is modified based on the given comment as Globally, there were 10 million people with TB in 2020, 1.2 million deaths from HIV-negative people and an additional 208,000 deaths from HIV-positive individuals.

Comments #3: Should the overall mortality rate be 3.7 per 100 person-years instead of 3.7 per person-years.

Author’s response: Yes indeed! It is amended based on the given comment

Comments #4: Should Debre Tabor compressive specialized hospital be Debre Tabor Comprehensive Specialized Hospital?

Author’s response: Sure! it has been modified based on the given comment

Comments #5: In “by taking sex as a predictor variable (on the male as the exposed group

denoted by q1 (0.38) and female group denoted by q0 (0.53)”, where the numbers come from?

Author's Response: Absolutely! we found this comment with the greatest relevance because it helps to convince readers who ask the same question after reading this paper.

The sample size can be determined by considering the previous study by taking its proportion/prevalence, incidence and associated/predictor factors. In this case, the sample size was determined based on the second objective (i.e., predictors. sex) from the previous study as we tried to cite after categorized as the probability of survival in exposed and non-exposed groups.

Comments #6: The sentence “The patients provided informed written consent to have data/samples from their medical records used in research.” needs re-wording as the patients were too young to provide such consent.

Author’s response: Accepted! In genuinely speaking, the informed written consent was obtained from ART focal person in each hospital on behalf of the study participants. Since it is secondary data. Hence, it is modified based on the given comment

Comments #7: In “half of 117 (51.54% and 122(53.74%) male and age between 6.10 years respectively.”, the order was wrong: 122 is for male and 117 for the age group.

Author’s response: Appreciated! Hence, it is revised based on the given comment!

Comments #8: In “118 (51.98%), 99 (43.61%), 103 (45.37%), and 218 (77.03%) …), what 218 is for?

Author’s response: Undoubtedly! it is removed since it is a typing error.

Reviewer #3: 

Comments #1: The authors studied lifetime survival and predictors of mortality in children with TB and HIV coinfection. The overall mortality rate was 3.7, which as associated with treatment failure, especially CD4 count below the threshold. The manuscript has improved adequately, but there are minor concerns that should be made clear for the conclusion in relation to treatment failure.

Author’s response: Great thanks for having this comment and concern in the following statements!

Comments # 2. Antiretroviral drug (ARV) combination in the patient cohorts and groups, especially those with treatment failure, since 18.94% co-infected children showed treatment failure;

Author’s response: We were categorized the study participants in a group at the very beginning of the study in the different predictor variables. In this case, the baseline of the study participants (i.e., co-infected children) regarding to treatment failure was grouped. and then we follow them up to the interesting outcome (i.e., death) development or censored.

Comments #3. Did the ART cessation happens in the co-infected patients, which likely contribute to treatment failure, such as CD4 decline to the levels of threshold and emergence of opportunistic infection. 

Author’s response: Sure! You know and we know that if TB and HIV co-infected children have treatment failure, the risk of CD4 decline and the emergence of opportunistic infection increased. 

In the meantime, CD4 decline and emergence of opportunistic infection can increase the risk of treatment failure. So that, we were categorized the study participant via different predictor variables of the dependent variable at the entrance of the study, which implies in controlling the confounding effect of one predictor variable on another predictor (the effect of treatment failure over CD4 count and OI to towards mortality). 

Comments #4. These are important to make statistical analysis and conclusion; The ARV combination and treatment duration in cohort with treatment failure should be considered in the group classification in the statistical analysis, at least discussed.

Author’s response: Appreciated! In this study, the effect of ARV combination (i.e. EFV-based

NVP, and PI) and treatment duration (<60 months vs ≥60 months) for the outcome variable was assessed/analyzed. However, it is not discussed and analysed with treatment failure. Hence, it is discussed the revised version manuscript in the discussion section that can be can be seen from in yellow highlighted manuscript.

---

## [Editor Report · Decision Letter 2]

17 Aug 2021

PONE-D-21-04696R2

Estimation of Lifetime Survival and Predictors of Mortality Among TB with HIV Co-infected Children After Test and Treat Strategies Launched in Northwest, Ethiopia, 2021; A Multicentre Historical Follow-up Study

PLOS ONE

Dear Dr. Chanie,

Thank you for submitting your manuscript to PLOS ONE. After careful consideration, we feel that it has merit but does not fully meet PLOS ONE’s publication criteria as it currently stands. Therefore, we invite you to submit a revised version of the manuscript that addresses the poor English writing, particularly in your discussion section.

We look forward to receiving your revised manuscript.

Kind regards,

Qigui Yu, M.D./Ph.D

Academic Editor

PLOS ONE

Journal Requirements:

Additional Editor Comments (if provided):

Suggest to seek professional English editing services to improve the flow and writing of your manuscript.
---

## [Author Response · Author response to Decision Letter 2]

26 Aug 2021

Point by point Author’s response to Editors and reviewers 

Editors

[General Comment] Please review your reference list to ensure that it is complete and correct. If you have cited papers that have been retracted, please include the rationale for doing so in the manuscript text, or remove these references and replace them with relevant current references. Any changes to the reference list should be mentioned in the rebuttal letter that accompanies your revised manuscript. If you need to cite a retracted article, indicate the article’s retracted status in the References list and also include a citation and full reference for the retraction notice.

Response: Thank you very much for the reminder. We have made revisions accordingly.

[General Comment] Suggest to seek professional English editing services to improve the flow and writing of your manuscript.

Response: In genuine speaking, we can’t obtain native English language experts to revised the manuscript intensively since we are living in highly resource limited setting. However, we have carefully considered the comments and tried our best to address every one of them to revised and to made the manuscript beauty. 

[General Comment] If reviewer comments were submitted as an attachment file, they will be attached to this email and accessible via the submission site. Please log into your account, locate the manuscript record, and check for the action link "View Attachments". If this link does not appear, there are no attachment files.]

Response: We are needless to state that we are happy to provide any more details in support of this manuscript as required.

We are observed the attachment file in action link, and there is one attachment file. However, all comments and suggestion in the attached file was addressed in the first round. Now, we are attached once ana again as follows. 

 Reviewer#1:

Comments #1: The major issue had been poor English writing, which was improved in this revised version. However, there are still many grammar mistakes and the overall flow of the sentences need to be fixed. There are also some misstatements and typos, some of which are listed below.

Author’s response: Undoubtedly! It is modified based on the given comment. 

After frequent proofreading of the manuscript had several grammatical wordings and spelling errors. Hence, based on the given comment revision was made. These changes are found throughout the manuscript. These changes are found throughout the revised version manuscript

Comments #2: The statement in the Abstract “However, there is no prior evidence in Ethiopia.” is not accurate as the authors themselves cited a reference to show the opposite (ref #18, Survival and predictors of mortality among children co-infected with tuberculosis and human immunodeficiency virus at University of Gondar Comprehensive Specialized Hospital, Northwest Ethiopia. A retrospective follow-up study. PLoS ONE [Internet]. 2018 May 22”). It appears that this current study and the above-mentioned 2018 study both recruited patients from the same Gondar Comprehensive Specialized Hospital. Were there any patients included in both studies? Some discussion on the similarities and differences between both studies would be helpful. As the risk of mortality is associated with CPT therapy non-users, the conclusion should emphasize CPT use along with ART therapy

Author’s response: We strongly agree with the essence of incorporating this comment!

Sure, we are stated that there is no prior evidence in Ethiopia regarding to estimation of lifetime survival and predictors of mortality among TB with HIV co-infected children after test and treat strategies launched in Northwest, Ethiopia, 2021. What makes this research differ from the previous one (i.e., ref #18 from 2005-2018) is: This study was conducted in after test and treat strategies launched (i.e., after 20014), recent evidence, and multicenter. However, we are concluded to revised as limited evidence than no evidence. 

Comments #3: Some of the references need to be updated. For example, World Health Organization. Global tuberculosis report 2020 is available (ref #4); The data shown in the sentences “In 2015 worldwide, 1.6 million people…” should be replaced with more recent data.

Author’s response: Unquestionably! It is modified based on the given comment as Globally, there were 10 million people with TB in 2020, 1.2 million deaths from HIV-negative people and an additional 208,000 deaths from HIV-positive individuals.

Comments #3: Should the overall mortality rate be 3.7 per 100 person-years instead of 3.7 per person-years.

Author’s response: Yes indeed! It is amended based on the given comment

Comments #4: Should Debre Tabor compressive specialized hospital be Debre Tabor Comprehensive Specialized Hospital?

Author’s response: Sure! it has been modified based on the given comment

Comments #5: In “by taking sex as a predictor variable (on the male as the exposed group

denoted by q1 (0.38) and female group denoted by q0 (0.53)”, where the numbers come from?

Author's Response: Absolutely! we found this comment with the greatest relevance because it helps to convince readers who ask the same question after reading this paper.

The sample size can be determined by considering the previous study by taking its proportion/prevalence, incidence and associated/predictor factors. In this case, the sample size was determined based on the second objective (i.e., predictors. sex) from the previous study as we tried to cite after categorized as the probability of survival in exposed and non-exposed groups.

Comments #6: The sentence “The patients provided informed written consent to have data/samples from their medical records used in research.” needs re-wording as the patients were too young to provide such consent.

Author’s response: Accepted! In genuinely speaking, the informed written consent was obtained from ART focal person in each hospital on behalf of the study participants. Since it is secondary data. Hence, it is modified based on the given comment

Comments #7: In “half of 117 (51.54% and 122(53.74%) male and age between 6.10 years respectively.”, the order was wrong: 122 is for male and 117 for the age group.

Author’s response: Appreciated! Hence, it is revised based on the given comment!

Comments #8: In “118 (51.98%), 99 (43.61%), 103 (45.37%), and 218 (77.03%) …), what 218 is for?

Author’s response: Undoubtedly! it is removed since it is a typing error.

---

## [Editor Report · Decision Letter 3]

11 Oct 2021

Estimation of Lifetime Survival and Predictors of Mortality Among TB with HIV Co-infected Children After Test and Treat Strategies Launched in Northwest, Ethiopia, 2021; A Multicentre Historical Follow-up Study

PONE-D-21-04696R3

Dear Dr. Ermias Sisay Chanie

We’re pleased to inform you that your manuscript has been judged scientifically suitable for publication and will be formally accepted for publication once it meets all outstanding technical requirements.

Kind regards,

Qigui Yu, M.D./Ph.D

Academic Editor

PLOS ONE
---

## [Editor Report · Acceptance letter]

19 Nov 2021

PONE-D-21-04696R3 

Estimation of Lifetime Survival and Predictors of Mortality Among TB with HIV Co-infected Children After Test and Treat Strategies Launched in Northwest, Ethiopia, 2021; A Multicentre Historical Follow-up Study 

Dear Dr. Chanie:

I'm pleased to inform you that your manuscript has been deemed suitable for publication in PLOS ONE. Congratulations! Your manuscript is now with our production department. 

Kind regards, 

on behalf of

Dr. Qigui Yu 

Academic Editor

PLOS ONE